# Educational Leadership Training, the Construction of Learning Communities. A Systematic Review

**Inmaculada García-Martínez** [1], **Miguel Á. Díaz-Delgado** [2] **and José Luis Ubago-Jiménez** [3,*]

1   Department of Didactics and School Organization, University of Granada, 18071 Granada, Spain; igmartinez@ugr.es
2   Institute of Research of the University and the Education, Autonomous National University of México (UNAM), Mexico City 04510, Mexico; miguelangel.diaz@comunidad.unam.mx
3   Department of Didactic of Musical, Plastic and Corporal Expression, University of Granada, 18071 Granada, Spain
*   Correspondence: jlubago@ugr.es; Tel.: +34-958-246-685

**Abstract:** Instructional leadership notions and practices allow educators to engage in relevant roles within schools. Instead of implementing these concepts in professional programs, Mexican and Spanish education systems still preserve a "technically oriented" training model that separates educational and professional aims. Diverse studies have identified the benefits of implementing instructional leadership orientations within "Educational cooperation", "Cooperative education", "Team teaching" and "Teacher leadership" at schools. This systematic review conducted using Web of Science—contributes by organizing the produced knowledge and identifies the main findings reported by the academic literature on this topic. It seeks to answer the following questions: (1) What are the contributions of this research to the education systems examined? (2) What kind of knowledge about educational leadership and professional learning communities can be inferred from them? Results from the majority of studies found that instructional leadership offers a useful tool to promote shared responsibility between teachers and head teachers and supports professional learning communities. A main conclusions of the present study is that it highlights the importance of bypassing existing bureaucratic practices within schools in order to replace the traditional "technical orientation" of training programs. Instructional leadership may facilitate some of the required transformations in the context of global educational reform.

**Keywords:** instructional leadership; teacher training; head teacher training; professional learning communities

## 1. Introduction

The area of educational leadership is commonly linked with the areas of educational administration and educational management. Traditional, at the beginning of the field of educational administration, which first emerged during the English Industrial Revolution, leadership was associated with achieving efficiency in educational institutions. This is different to the managerial perspectives, which is specially raised in the context of the global economy and considers leadership as a process for improvement within educational organizations.

Leadership is conceived as a situation of superiority within an institution or organisation that delegates to the leader the power to direct a group which is responsible for and has a view to achieving previously defined objectives. Visibly, the leader heads the organisation and exerts its image. In this way, specific power dynamics emerge between the members of the organisation managed by the denominated leader (Harris 2014; Piot and Kelchtermans 2016).

In this way, leadership must be implemented from at the heart of the school, being considered as a quality that includes the educational community (Gronn 2002; Gronn 2009). The institution as an organisation framework, in which educational agents coexist and interact, demands a distributed leadership throughout the organisation (Göksoy 2015; Hulpia et al. 2012; Hulpia et al. 2009; Spillane 2006; Spillane 2013).

It has been argued that when a series of conditions are established that lead towards collaboration (Bahous et al. 2016) and the vision of the centre is redirected towards pedagogy, improvements in student learning are produced (Gumuseli and Eryilmaz 2011) and leadership becomes oriented towards pedagogical dynamics which are distributed in the pursuit of an organisational transformation (Harris and Jones 2010). Furthermore, educational leadership can be conceived in two different senses. Firstly, it can be considered as a set of relationships (Spillane et al. 2004) where leadership capacity is seen as a process of inspiration, since it emphasizes the transmission and acceptance of a vision. Or, it can be considered in the sense of (Lazaridou and Beca 2015) achieving educational goals and those set in a research area focused on the study of the dynamics between educational structures and the interests of stakeholders. In both cases, educational leadership research identifies three main concepts: access to decision making, agency of stakeholders, and organizational structure of educational systems.

This area of practice and research holds great importance for the improvement of professional competencies in educational fields, providing notions of shared responsibility and participation, as well as engendering synergies between teachers and other members of the educational community (Leithwood and Duke 1999).

Within the framework of global educational reforms, professional teacher training is identified to be one of the pillars for the improvement of educational leadership. It has been assumed that teachers and head teachers are losing professional autonomy, a situation that discourages individuals from following paths of re-professionalization where they are committed to raise their pedagogical, social, organizational, technical, evaluative and ethical competency levels (Li et al. 2016).

Professional training as a permanent process nurtured by educational leadership contributes to improvements in the design and development of teaching and learning processes (Marcelo 2012). "It is now widely accepted that teachers need initial training to be effective classroom practitioners and continuing professional development throughout their careers" (Elmore 2010, p. 418). According to educational leadership theories, especially those linked to instructional perspectives (Bush and Jackson 2002), schools play a vital role by providing a permanent professional learning space.

This article critically analyses how teacher education is understood from a "technically oriented" perspective in some global education systems, particularly in Spain and Mexico. In considering an alternative educational leadership training model for teachers and head teachers oriented towards professional learning communities, the present study reports a systematic review with the aim of providing a succinct theoretical understanding in which teaching and school management converge in the professionalization of school leadership.

Teacher and head teacher training is indispensable for the improvement of educational systems as professional training programs for leadership within teachers and head teachers may develop notional and applied bases. However, the traditional "technical orientation" of training boosted by technocratic education reforms usually separates teachers and head teachers when undertaking training. This results in them having very differentiated functions and can even produce a degree of antagonism with regards to their understanding of their development in the educational system. This serves to diminish the impact of cooperative learning in educational community practices. When this perspective is taken, training is clearly understood as "an event" where teachers and head teachers are called upon to improve separately, instead of a process where they may learn and apply their new knowledge together in a community of learning.

Educational systems in both Spain and Mexico are now finding common challenges in teacher training and the connection with the communities they attend. In both systems "the [social] context acquires more and more importance, [as well as] the ability to adapt to it methodologically, the vision

of teaching not so much technical, as the transmission of a finished and formal knowledge, but rather as a knowledge in construction and not immutable" (Gawlik 2018, p. 4). Teachers and head teachers from the public education systems within these two countries are facing unstable social, cultural and economic challenges, which deeply influence the dynamics of school communities and newer "technically oriented" training programs have been introduced in response to these social scenarios.

Nevertheless, the "technically oriented" training model for teachers and head teachers seems to be more focused on functions rather than organizational objectives, following bureaucratic routines that fail to solve the current problems in schools and communities. Additionally, teachers and head teachers in this model concurrently tech on programs deliver a professional service, commonly these training programs do not connect leadership theory and school cooperative learning practices. Such factors disturb teacher practices and management functions.

"Technically oriented" training programs duplicate school duties because professionals, instead of finding solutions together in learning communities, are individually focused. The opportunity of linking training programs and everyday duties in schools is latent, but the "technically oriented" training programs commonly do not exploit these opportunities.

It is important to change models of training programs in educational leadership. Albeit, true disciplinary knowledge is needed to develop in a curricular path, it is also required to undertake activities for a comprehensive educational and contextual approach. Educational leadership, as a set of theories focused on decision-making and shared responsibility among agents of the educational system, seeks for teachers and head teachers to progressively share responsibilities in addition to a pivot of practices for educational improvement (Imbernón 2002). One of the responsibilities in this relationship is the professional training itself.

Global education systems, but particularly Spanish and Mexican ones, are hypothetically considered as prototypes of post-bureaucratic reorganization (Esteve 2009). Using this perspective instructional leadership is privileged and the position of a "leader" is more important than administration. This position is not limited to head teacher duties in this category of education systems and it is feasible to build foundations within the school community for collaborative organizational development (Bolívar 2015). Instructional leadership in a learning community of practice requires motivation of school staff, encouragement and involving all educational agents to achieve education goals.

Educational leadership theories, especially those grounded in instructional perspectives, ultimately encourage collaboration in schools and the linking of teaching and head teacher oversight by removing classroom, school and community barriers. Nonetheless, it is important to demonstrate how leadership in education can help in reaching this aim. Thus, the present systematic review provides evidence of the inter-relatedness between the research areas of educational leadership, professional communities of practice and cooperative learning.

In an era of renovation of the education systems, it is necessary to analyze the organizational reconfiguration of schools and we consider that it is also important a transform school professions via the training programs they provide. To this end, the academic literature analysing instructional leadership at different levels is examined.

The perspectives of instructional leadership ease the understanding of distributive models of responsibility, this characteristic can reorient the practices of the community, offering them a continuous sense of learning. Instructional leadership within teachers plays an essential role in achieving expectations for student outcomes and in recognizing their impact on the organization and promotion of commitment to professional learning opportunities (Bennis 2000).

Instructional leadership has implications for system quality practices, as well as for competencies in school communities. These different connotations must be considered when designing training programs for school staff, in which head teachers and teachers are considered as key positions due to their direct contact with teaching, with the students and with the context. Training programs for educational leadership should not ignore that "it is no longer enough to teach what the curricular

framework has defined [...] teachers need to demonstrate that their students know how to think, solve problems, search and synthesize information, at the same time being able to face their learning autonomously and in collaboration with others" (Timperley 2008, p. 107).

An instructional approach to leadership training is also required due to the demands of the context and global society. Innovative professional practices based on instructional leadership that seek to update or replace obsolete techniques and training programs can help teachers and head teachers to rethink their role within the learning community. According to the examined academic literature, professional learning communities and cooperative learning may contribute to shifts in current practices based on accountability of head teachers and basic didactic relationships between teachers in classrooms.

Training oriented through instructional leadership theories focused on school communities can facilitate to develop competencies based on pedagogical knowledge, ethical and moral commitment, and co-responsibility for organizational objectives and challenges.

Instructional leadership competencies and its theoretical framework are unfortunately not sufficiently contemplated in the process of teacher training from the Spanish and the Mexican training programs, even assuming that Teachers and Principals are "an essential factor in educational quality and that there is a need to offer them initial and permanent professional training that allows them to rise to the challenge" (Montecinos 2003, p. 105).

From the research projects congregated in this study, it has been detected that Teachers and Principals' training programs in both education systems are in fact experiencing [a] deep differences between the novel teachers' role with less than 5 years in service and the experienced teacher, [b] a conceptual and practical separation between the knowledge of Principals and Teachers, even when they are part of the same schools, [c] low significance of the learning that promotes the training programs to meet school goals, [d] a professional isolation that acutely segregates principalship from teaching (Díaz 2018).

However, teacher training is nowadays experiencing a transition from the "technically oriented" model to a model of professional community of practice, characterized by the sample of multiple conceptions of education, that highlight an integral vision of education, focused on a greater attention to students' diversity and the promotion of interdisciplinary experiences, confrontation of ideas and participation, in other words, education leadership.

Professional training understood as a space of (Bush and Jackson 2002) "development where teachers [and head teachers] can systematically attempting to improve [their] professional practice, beliefs and professional knowledge, with the purpose of increasing the quality of teaching, research and management" (p. 19). Herein the apprenticeships between initial and continuous training, recognize the daily experience in the classroom, the socialization of the professional practice, the expectations for improvement in learning methods and techniques, the school environment, as well as possibilities for career advancement and involvement in the teaching work of the different hierarchical structures of the education system. This is process is the expression of a learning community of practices.

Professional communities of practice are inspired by the instructional perspectives of leadership, where relationship between Teachers and Principals point to the (Contreras 2016) "the transformation of the education and the development of the school it is essential that Teachers and Head teachers be duly trained and empowered, and that they become aware of their leading role in these processes" (p. 232) more than privileging of hierarchies in schools.

From the view of supranational educational policies, a common denominator is seen: the formal discourse of the educational system calls for the empowerment of Teachers and Head teachers, to assume an instructional leadership role, however, the preservation of schemes bureaucratic impedes it. This reforms discourse appears to pursue a reorientation of the practice of Head teachers and Teachers to transcends their function borders and generate changes at the community level.

Mulford (2006) affirms that the keys to the development of the center-community relationship lie on (a) the need for a clearer commitment of the Head teacher at the community; (b) the requirement of

dear Principal commitment to the school and community; (c) the presence of an active search eagerness involving all sectors of the community; (d) a higher sensitivity in the management of school-community agreements; (e) the empowerment of diverse stakeholders in decision making, including teachers; (f) extending internal and external networks between the school and the community; (g) seeking a shared vision of future mainly focused on the development of the youngest; (h) the opening, both of the school and the community, to new ideas, risks and opportunities that arise; (i) pursuing an active role in the whole school and the community from all the stakeholders; (j) the assessment of all the skills that contribute to individual, school and community learning; (k) the conception of leadership as a collective responsibility; and (l) the acceptance of the school as a learning community.

In the aforementioned keys, instructional leadership is a transversal characteristic for the Teachers to take part on a relevant role in community learning practice, but contrary to incentive this capacity from training programs, their processes hardly recognize conceptually and procedurally, the regenerative condition of teacher training in leadership.

Kilpatrick et al. (2001) establish that the perspectives of educational leadership can provide motivation and sustainability to both, the school and community stakeholders. The function of educational leadership can generate relationships between the school and the community in which it is inserted.

Education systems have not surpassed yet the traditional training schemes, that separate the Teachers from Principals, what is more, the "technically oriented" divides them towards the resolution of problems of the school and the school community.

It is required to find the studies that reaffirm educational leadership, as a reference for the learning of Teachers and Head teachers that answer the questions: what are the contributions of such research to the education systems in question? and what kind of knowledge about educational leadership and professional learning communities can be inferred from them?

In order to answer these questions, we depart from the assumption of that it is necessary to develop conceptual and procedural comprehension schemes on Teacher and Principals training in educational leadership. For this, we developed a systematic review that contributes to extending knowledge about the conjunction between Teacher training and Head teachers in educational leadership.

## 2. Materials and Methods

A systematic review contributes to identify the main ideas found in the academic literature on a particular matter (Cardoso et al. 2014), for this systematic review, we followed the guidelines from the PRISMA declaration (Liberati et al. 2009); searching the notions of instructional leadership within Teacher and Principals training programs, and the intersection of the concepts of cooperation, collaboration, teaching teams.

### 2.1. Procedure

As said, we analyzed the studies that link Teachers', Principals' training, and instructional leadership. It was developed an exhaustive search in the Web of Science (WOS) database, inputting the following keywords: "Educational cooperation", "Cooperative education", "Team teaching" and "Teacher leadership" all of those concepts linked to instructional leadership theories (Table 1). The range of found writings was reduced by pondering just the articles published between 2007 and 2017, obtaining a result of 283 (Figure 1).

**Table 1.** Sample selection process.

| Initial Search |
|---|
| "Educational cooperation", "cooperative education", "team teaching", "secondary schools" y "teacher leadership" |
| Results found |
| WOS: 283 articles |
| Duplicates |
| 97 articles |
| Inclusion criteria |
| Only articles: "Social Sciences", "Education Educational and research", *n* = 68 |
| Final Selection |
| Empirical articles that relate leadership and pedagogical coordination, participants, professors or headteachers. *n* = 21 |

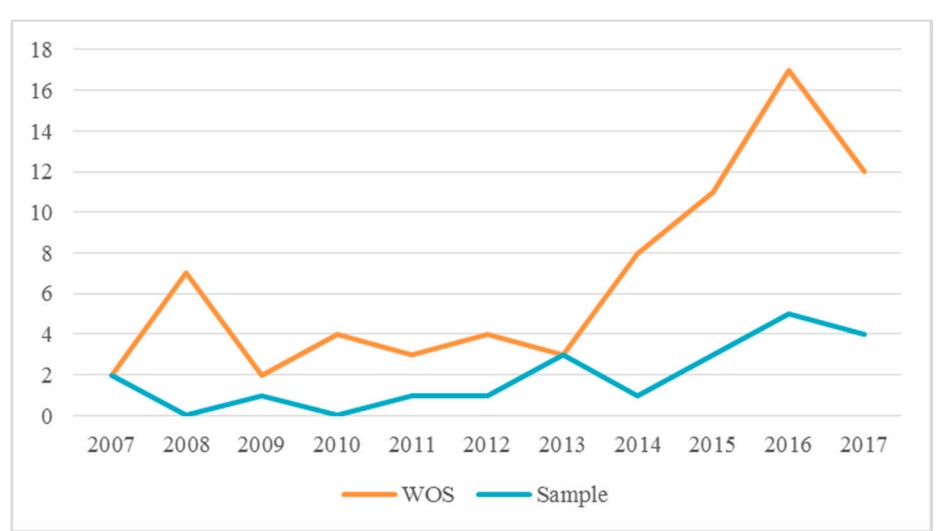

**Figure 1.** Distribution of publications from 2007 to 2017.

To deeply refining the search, there were considered just those academic articles from the field of "Social Sciences" and, to delimit the number of articles, there were included papers written in English and Spanish language. A final filter was included, decreasing the number of publications to those included in the "Education Educational Research".

The initial reading of the selected articles was focused on the title and summary of the paper. In the following phase, the reading was centered on the method, results and conclusions of the research's, bearing in mind the relevance and co-relation of categories found. The procedure concluded with a thorough reading of the pre-selected full texts, and finished with the selection of 21 articles that constitute the present research. The data was processed materialized by comparing the logical order of the information, synthesizing all the information obtained in order to reach the achievement of a relevant and truthful study.

## 2.2. Sample

The procedure outlined above resulted in a population of 283 articles from the Web of Science database. Finally, after applying the filters, the sample was established in 21 articles.

## 3. Results

*Data from Studies Selected for Systematic Review*

The 21 articles of this systematic review are characterized by being based on qualitative research, this leads to the sample participation (although some studies do not indicate the number of the sample), be around 5133 (Table 2). To make visible the most significant characteristics of the articles, we propose the next table:

**Table 2.** List of articles that make up the systematic review.

| Author | Year | Population * | Sample | Instruments * |
| --- | --- | --- | --- | --- |
| Cravens & Drake | 2017 | S, U | 27 schools | O & I |
| Fung & Lui | 2016 | S | 152 students | Q & I |
| Feito | 2013 | S | Students & teachers | PO, I & DG |
| Androniceanua, Risteaa & Mascu Uda | 2015 | S | 96 teachers | Q |
| Opdenakker & Van Damme | 2007 | S | Students & teachers | Qualitative method |
| Li, Hallinger & Ko | 2016 | P | 32 schools | Cross-sectional research & quantitative methods |
| Sans-Martín, Guàrdia-Olmos & Triadó-Ivern | 2016 | P, S | 3.835 schools | Indicators of the TALIS study. |
| Mendo-Lázaro, Polo-del-Río, Iglesias-Gallego, Felipe-Castaño & León-del-Barco | 2017 | U | 750 students | Q |
| Larraz, Vázquez & Liesa | 2017 | U | 127 students | Q |
| Dyson, Colby & Barratt | 2016 | P, S | 12 teachers | CS |
| Cohen & Zach | 2013 | U | - | Q |
| White, Wertheim, Freeman & Trinder | 2014 | P, S | 13 schools | CS |
| Dochy, Berghmans, Kyndt & Baeten | 2011 | P | - | R |
| Sancho-Thomas, Fuentes-Fernández & Fernández-Manjón | 2009 | U | 835 students | NUCLEO program: CS & Q |
| Mellado-Hernández, Chaucono-Catrinao, Hueche-Oñate & Aravena-Kennigs | 2017 | P, S | 36 teachers | CS |
| Oddone | 2016 | S | 47 teachers | CS |
| Goodyear & Casey | 2015 | S | 6 teachers | I |
| Park, Kim, Park, Park & Jeong | 2015 | S | 3 science teachers & 3 middle schools | POCoM & KTOP |
| Lin, Hong, Yang & Lee | 2013 | P, S | 91 students & teachers | PO; I & DG |
| Biasutti | 2012 | P, S | 177 teachers | Q |
| Lee | 2007 | S | 80 teachers | Q |

* U: University; S: Secondary education; P: Primary education; E: Early education; R: Revision; CS: Cases studies; I: Interviews; PO: Participant Observation; O: observations; Q: Questionnaires; DG: Discussion group; TALIS: Teaching and Learning International Survey.

## 4. Discussion

The profound and reflective reading of the above-mentioned papers foretastes the instructional leadership research in different educational scenarios in the post-bureaucratic age of educational leadership. The found permitted to assume that from the perspectives of educational leadership there is the possibility of generating co-responsibility between Teachers and Principals, primarily because both converge in specific educational communities and share specific and shared goals. It is established the need for a new way of understanding Teacher and Principals´ training that implies a redistribution of school responsibilities since more parallel schemes.

Androniceanua et al. (2015) affirm that the transformation and regeneration of the school is founded on a self-management system, decentralizing teaching and professionalizing at the same time the school leadership, beginning deeper cooperation in common projects. A modern education systems need to develop teaching leadership skills to ensure project-based management, especially focused in solving problems of schools and school communities. This found concurs with Biasutti (2012) who recognized the relationships between factors, such as innovation or teacher cooperation, on improving education.

Opdenakker and Van Damme (2007) emphasized the impact of the school context, the characteristics of the students and the grade of development of Teacher leadership competencies at a Secondary school in Flanders. This study revealed that the characteristics of the school and the degree of cooperation among its stakeholders have a positive influence on teaching; certainly, these factors are the most important to determine the success of learning. Hence, we conclude that context is a substantial element in the understanding of educational leadership processes. Also, teacher training in leadership must be envisioned and addressed instrumentally, methodologically and conceptually. Mellado-Hernández et al. (2017) found that cooperative learning and ethical and social action are essential to achieve contextual pedagogic practices; they demand to develop an inclusive curriculum, with emphasis on the didactic dimensions. Fung and Lui (2016) corresponded those found by contrasting the positive effects of teachers' cooperative work on student achievement.

Instructional leadership is derived from school cooperation, this, as a secondary effect removes hierarchies between Teachers and Principals. Li et al. (2016) found different "paths" for leadership that attempt to influence teaching and learning. This process can take different routes or objectives, addressing trust, cooperation, communication, guidance and support for students, linking, coherence and structure, teachers learn from parallel structures and instructional school environments.

Park et al. (2015); Sans-Martín et al. (2016) considered that cooperation and conversation of practices are deeply related to leadership styles. They found primarily two main styles of leadership; distributed and instructional. These styles of leadership are dependent on several factors, such as the organization of human resources in each school, the characteristics of the students and their need for training, the degree of cooperation and collaboration among the teachers, driven by the direction in the implementation of academic activities and the involvement professional collaboration.

We conclude that the designing of Teacher and Principals´ training programs should consider leadership styles, centred in a contextual analysis and a pluralistic knowledge of the theoretical perspectives; in this regard, it is needed that Teachers and Principals could recognize and implement the leadership style that contributes most to the context of practice.

Feito (2013) states that succeeding in implementation of changes in schools depends on the goodwill of the school staff, and warns that different impediments may arise, such as the irrevocability of some members of the school to the idea that it can work better in a different way. This is usually motivated by a sector of traditional teachers who prefer to keep a passive role, which almost forces them to forget their interests and motivations. Teacher training in educational leadership must assume these obstacles will be presented soon or later, and that it is necessary to facilitate the collaboration even with those teachers that show lack of motivation. In this aim, other teachers have much to do, specifically because when a Principal stablishes tries to find new ways to achieve same goals, authority

acts could arise, and if is a colleague who enables the partnership and gives the guidelines for action, this could motivate others to follow this path.

Cravens and Drake (2017) propose that, in order to impulse a pedagogical transformation in schools, it is necessary to implement a model of teacher peer excellence groups (TPEG). This technique comprehends school staff organized by subject or educational levels, fully involved in learning communities and in improving teaching. Teacher training should seek decentralization, abandon the idea of an exclusively technical training of elite training staff and landing on the support from an efficient feedback, collaborative planning and the promotion of co-responsibility. This process is viable in there is some room for shadowing between Teachers and Principals, especially if the bureaucratic tasks in the spaces of School Technical Council (CTE) in the Mexican and the Teachers Cloister at Spanish educational systems are reduced to the maximum.

The professional challenges of a permanent staff training, the opening to new information, and the acquisition of new knowledge implies the implementation of newer technologies. Coto et al. (2016) highlighted "u-learning" or "ubiquitous learning" like fundamental model for school communities, and argued that this can actually an opportunity for training to extend the time and space possibilities. They specified that this training modality of instructional leadership should include collaboration, impulses communication skills, foster cooperation and team-working.

In Sancho-Thomas et al. (2009) is studied the implementation of an electronic learning framework (NUCLEO). Indeed, their design promotes effective cooperation and skills for team-working, also promoting the active participation of stakeholders in the teaching/learning process. Lee (2007) requests the adoption of a collaborative approach as a strategy to improve teaching practices. Similar results were obtained by Oddone (2016). He examined information technology and communication in the classroom, as an axis for training and how these affect the familiarity of Italian secondary school teachers with ICTs, active learning and cooperative work.

Mendo-Lázaro et al. (2017) identify some factors that promotes school improvement. It is evidenced in a better understanding of attitudes and preferences regarding cooperative learning and collaboration in schools.

Also, Lin et al. (2013) analyzed the impact of collaborative and cooperative work on professional development, accounting the benefits they report for the improvement of practices and student learning.

Larraz et al. (2017) analyzed the effects of cooperative learning; the results show the development of transferable skills, such as leadership skills, negotiation, reflection, teamwork, improve social interactions. Therefore, cooperative learning can be considered a key instrument for the professionalization of Teachers and Principals in training programs.

Dyson et al. (2016) studied the implementation of a Cooperative Learning model (CL) at the primary level. They obtained data from interviews, field notes and analysis of certain documents and its analysis derived in identified insufficient training of the Ed. Physics teachers in instructional leadership, in improving social skills to enhance cooperative learning, a lack of awareness on the part of teachers about cooperative learning and in skills to pedagogically orientate students. The study states the need for an instructional shift of the training, which includes the colleagues themselves, as "critical friends" in the school context. Cohen and Zach (2013) add to this conclusion, that instructional leadership notions can contribute to cooperative learning and this factor had a special impact on the effectiveness of teaching and planning of teaching.

Within the area of Ed. Physics, Goodyear and Casey (2015) show the difficulties that have some shadowing techniques in educational innovation. They searched what were the keys that allowed introducing the pedagogical changes and new models of leadership.

The school understood as a learning community space helps to consolidate training programs, where the resolution of conflicts must be both a priority and an aspect to be prevented. White et al. (2014) promote a team model characterized by the support of the educational community in solving problems; an innovative intervention model for intervening specific problems of schools, finding to conjugate these occasions with the possibilities learning in situations methods. They exposed how

co-responsibility is developed by this instructional leadership practices and this accurately brought shared leadership, encouraged by the Principals.

Dochy et al. (2011) analyzed the instructional principles for achieving learning communities, and exposed certain obstacles and common barriers that schools must face during this process. This makes us think that the educational reforms sustained in school improvement perspectives imply a shift from the uniqueness in the "technically oriented" Teacher and Principals training programs to a comprehensive model of training in practice, that could impulse authentic professional learning communities in schools, which will involve the recompositing of the dynamics centered in the institutional bureaucracy, towards an instructional model.

## 5. Conclusions

This study explores critically the procedures still focused on the bureaucracy present in the Spanish and Mexican educational system despite the efforts of the current educational reforms in both countries; additionally, it was stressed how both education systems separate Principals and Teachers in their training, even recognizing the importance of school based learning. These "technically oriented" models openly differentiated learning objectives for Teachers and for Principals that does not match with school community learning practices.

However, training programs in question may address the understanding of the school communities, the objectives and training needs of both functions have broad similarities and lie in the achievement of institutional goals and the resolution of problems of the educational community, both inexhaustible sources of experiences for professional learning from instructional leadership.

The 21 articles found in this systematic review showed how instructional leadership notions may provide procedures for construction learning communities within the schools, therefore Teacher and Principals´ training can be in this way extending their effects to permanent learning contexts.

Several studies were found in primary and secondary schools, constituting the primary level of education, but also some developed at the tertiary level, specifically in Universities. It is important for the studies that converged in this paper to consider that educational leadership is not an exclusive element of school Principalship, especially if the instructional models consider teacher involvement and cooperation as an essential matter.

It is also important to consider that Teacher training is not just an "event" with a specific beginning and a deadline, it is more a process for enable continued learning. It is important that, for the Spanish and Mexican school systems to regenerated principal and teachers´ programs, allowing the observation of logics of school learning communities.

Most of the articles found in this review asserted on the need for the establishment of a culture of teacher collaboration and the predominance of distributed systems of leadership based on instructional aims. They showed that teacher-principal collaboration could contribute to develop a professional learning community.

The studies found argue that collaboration within educational organizations is also a feature of teaching professionalism and therefore a continuous training element. The bureaucratic dynamics that prevail in both education systems can be gradually shifted to a model focused on instructional leadership. This is the reason why it seems necessary to drastically reduce bureaucratic demands to the school.

The school is an excellent learning community even for professionals. Instructional leadership orientation within the relationship of Teachers and Principals in training programs can promote the permanent link between theory and practice to help stakeholders to better understand and visualize goals and challenges.

The training programs for Teachers and Principals can provide of certain guidance of instructional leadership to foster collaborative practices and to propels systemic change to consolidate a sustainable and impassive improvement at the time.

**Author Contributions:** I.G.-M. and J.L.U.-J. conceived the hypothesis of this study. M.Á.D.-D., I.G.-M. and J.L.U.-J. participated in data collection. J.L.U.-J. and I.G.-M. analysed the data. All authors contributed to data interpretation of statistical analysis. M.Á.D.-D. and J.L.U.-J. wrote the paper with significant input from I.G.-M. All authors read and approved the final manuscript.

**Funding:** This research received no external funding.

**Conflicts of Interest:** The authors declare no conflicts of interest.

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
