# Peer review of "Educational Leadership Training, the Construction of Learning Communities. A Systematic Review"

_socsci, doi:10.3390/socsci7120267_

Round 1

Reviewer 1 Report

Though not innovative, the subject is significant for the training of education professionals and in general to improve the quality of the present day school system.

The concepts expressed and the data extracted from the analysis of the literature are of a more generic nature and therefore transferable to other contexts.

The ideas employed and the considerations on the topic prove to be of great interest and offer practical applications (e.g. lines 81- 84),  but could be better defined and potentially integrated with further references to the literature.

For example, it would be advisable to examine more in depth the concept of educational leadership in the introduction (line 28), also from a hystorical point of view, and illustrate the relationship between distributive perspectives of  leadership and cooperative learning, and between educational leadership, professional communities of practice and the development of the school (and students’ learning) as reported in the text on several occasions.

Referring to the multidimensional nature of leadership (lines 86-88)  also proves interesting and this aspect should be developed more deeply.

From a methodological  point of view, it would be beneficial to clarify the choices made when analyzing the literature (e.g. criteria and process of analysis and description). In paragraphs 3 and 4, for instance, certain key-words and literal quotations could be reported.

It might be useful to restructure the introductory section, in  order to simplify  the understanding of the topics and research stages involved.

The broad introduction could be divided into a first section which includes  a general illustration of the matter and an identification of the specific issue (lines 155-158), followed by a second part defining the objectives  set by the authors (lines 37- 38; 46-47).

Formal corrections:

-       please integrate and clarify all references to pages in brackets (Lines: 128; 139;…)

-       check spelling in the text (e.g. line 30 missing final “s” in “generate”)

Author Response

Suggestion 

   Attended?

Main changes in   lines 

The concept of   educational leadership in the introduction (line 28), also from a historical   point of view

The concept of   educational leadership was extended, relation it with an historical view in   its relationship with educational administration and management

27- 44

The relationship   between distributive perspectives of leadership and cooperative learning, and   between educational leadership, professional communities of practice

It was stated   the instructional leadership has effects on distributive practices at the   communities of practice, specially on professional ones. 

113-122,

144-149,   184-185, 210-212, 301-311.

Introduction could be divided   into a first section which includes a general illustration of the matter and   an identification of the specific issue (lines 155-158), followed by a second   part defining the objectives set by the authors (lines 37- 38; 46-47).

It was edited   according to the suggestion

5-22

Please integrate   and clarify all references to pages in brackets (Lines: 128; 139;…)

It was edited   according to the suggestion and all references have been clarified

check spelling   in the text (e.g. line 30 missing final “s” in “generate”).

It was edited   according to the suggestion

Reviewer 2 Report

It is necessary that in section 2.1. Procedure, include in greater detail the final selection of the 21 articles. The search strategy should appear in WOS and SCOPUS, step by step, in order to find it.

Author Response

Suggestion 

   Attended?

Main changes in   lines 

Section 2.1.   Procedure, include in greater detail the final selection of the 21 articles

In the section   "2.1 procedure" the steps followed in the search for manuscripts   are detailed step by step. The sample selection process is also detailed. In   addition to this point, the selection criteria are specified in Table 1.   Indicating the process for obtaining the 21 selected articles.

243-254

Table 1

The search strategy should appear in WOS and SCOPUS, step by step, in   order to find it.

Our search has   been based on Web of science, as we consider it to be the main database of   the social sciences area (JCR). In other studies we will also consider   SCOPUS.

Reviewer 3 Report

The standard of English made it impossible to detect the argument made in this article.I am therefore sorry I cannot give a detailed review. From what I could discern there appeared to be no alignment between the title, the abstract and the content. It is evident a systematic literature review took place. However the search terms did not appear to link with the tenor of the article. They were:“Educational cooperation”, “cooperative education”, “team teaching”, “secondary schools” “teacher leadership" Those chosen 'related to leadership and pedagogical coordination, participant, professors or headteachers' (Table p.5)  in an article titled Educational leadership training. Was it about evaluation of training programmes for leaders and teachers or about pedagogical leadership to support professional learning communities? There are possibly interesting findings here that need to be expressed in quality English and in a coherent manner

Author Response

Suggestion 

   Attended?

Main changes in   lines 

The standard of   English made it impossible to detect the argument made in this article

We made a   greater edition all long the writing

All the writing

They were: “Educational   cooperation”, “cooperative education”, “team teaching”, “secondary   schools” “teacher leadership" Those chosen 'related to leadership   and pedagogical coordination, participant, professors or headteachers' (Table   p.5) in an article titled Educational leadership training.

We linked the   concepts of instructional leadership with others, the writing was mainly   focused on finding instructional leadership research that exposed effects on   distributive practices (cooperative, team teaching, teacher leadership) and   we conclude this notion of educational leadership could contribute in   training for teachers and principals to erase barriers and transforming   schools in professional learning communities; therefore we developed a   systematic review to sustenance our hypothesis 

 113-122, 133-136,

144-149,   184-185, 210-212, 310-311,

335-342

There are possibly   interesting findings here that need to be expressed in quality English and in   a coherent manner

Yes, we edited   the writing to improve the drafting 

All the writing

Round 2

Reviewer 2 Report

De acuerdo con la justificación, es suficiente.

Author Response

Thank you very much for your contributions to improve the manuscript.

Reviewer 3 Report

I appreciate considerable work has been undertaken to improve the English. Unfortunately  it is still difficult to understand. There are glimpses of the argument but there is not a consistent standard of English to grasp the overall focus. From what I could gather there are three threads:
Instructional leadership can foster professional learning communities

Teacher training should include collaborative teaching/leading

Collaboration would improve the educational system

However none are fully developed.

This was a succinct paragraph that could have been the focus of the article

Most of the articles found in this review asserted on the need for the establishment of a culture of teacher collaboration and the predominance of distributed systems of leadership based on instructional aims. They showed that teacher-principal collaboration could contribute to develop a professional learning community.

In addition there needed to be:

A definition of instructional leadership,professional learning communities, team teaching, teacher leadership

References for the claim that teacher education is technically based

Synthesis and integration of the literature review findings

Author Response

Thank you again for your contributions to improve the manuscript.

We have listened to all your recommendations and we have also made an English proofreading.

We have also added the definitions of educational leadership, professional learning communities, team teaching and teacher leadership.

In addition, more references have been added to support these definitions and bring greater scientific rigour to the manuscript.

Again, thank you very much for your contributions and we hope that we have met your requests.